# BOUNCE AND LEARN: MODELING SCENE DYNAMICS WITH REAL-WORLD BOUNCES

**Senthil Purushwalkam**[*] **& Abhinav Gupta**
Robotics Institute, Carnegie Mellon University
{spurushw,abhinavg}@cs.cmu.edu

**Danny Kaufman & Bryan Russell**
Adobe Research
{kaufman,brussell}@adobe.com

## ABSTRACT

We introduce an approach to model surface properties governing bounces in every-day scenes. Our model learns end-to-end, starting from sensor inputs, to predict post-bounce trajectories and infer two underlying physical properties that govern bouncing - restitution and effective collision normals. Our model, Bounce and Learn, comprises two modules – a Physics Inference Module (PIM) and a Visual Inference Module (VIM). VIM learns to infer physical parameters for locations in a scene given a single still image, while PIM learns to model physical interactions for the prediction task given physical parameters and observed pre-collision 3D trajectories. To achieve our results, we introduce the Bounce Dataset comprising 5K RGB-D videos of bouncing trajectories of a foam ball to probe surfaces of varying shapes and materials in everyday scenes including homes and offices. Our proposed model learns from our collected dataset of real-world bounces and is bootstrapped with additional information from simple physics simulations. We show on our newly collected dataset that our model out-performs baselines, including trajectory fitting with Newtonian physics, in predicting post-bounce trajectories and inferring physical properties of a scene.

## 1 INTRODUCTION

Consider the scenario depicted in Figure 1. Here, a ball has been tossed into an everyday scene and is about to make contact with a sofa. What will happen next? In this paper, we seek a system that learns to predict the future after an object makes contact with, or *bounces* off, everyday surfaces, such as sofas, beds, and walls. The ability for a system to make such predictions will allow applications in augmented reality and robotics, such as compositing a dynamic virtual object into a video or allowing an agent to react to real-world bounces in everyday environments.

We begin by observing that humans exploit both visual recognition *and* direct physical interactions to estimate the physical properties of objects in the world around them. By learning from a large number of physical interactions in the real world, we develop an approximate visual mapping for physical properties (e.g., sofas are soft, tables are hard, etc). However, two surfaces that look the same may produce vastly different outcomes when objects are dropped upon them (e.g., query for "happy ball and sad ball" on YouTube). Without observing these interactions, we would have no way of knowing that the surfaces are made of materials with differing physical properties. Motivated by this observation, in this paper, we investigate the application of physical interactions to probe surfaces in real-world environments, infer physical properties, and to leverage the interactions as supervision to learn an appearance-based estimator. Leveraging the regularity of spherical collisions, we adopt a simple ball as our probe. Our goal is to use captured probe collision trajectories to predict post-bounce trajectories and estimate surface-varying coefficients of restitution (COR) and effective collision normals over complex, everyday objects.

Rigid-body physics has often been employed to model collision events (Bhat et al., 2002; Brubaker et al., 2009; Kyriazis et al., 2011; Monszpart et al., 2016). However, real-world objects deform under collision, and so violate rigid-body assumptions. Collision normals and COR model a complex pro-

---

[*]Work done at Adobe Research during summer internship. Dataset and code available here: http://www.cs.cmu.edu/~spurushw/projects/bouncelearn.html

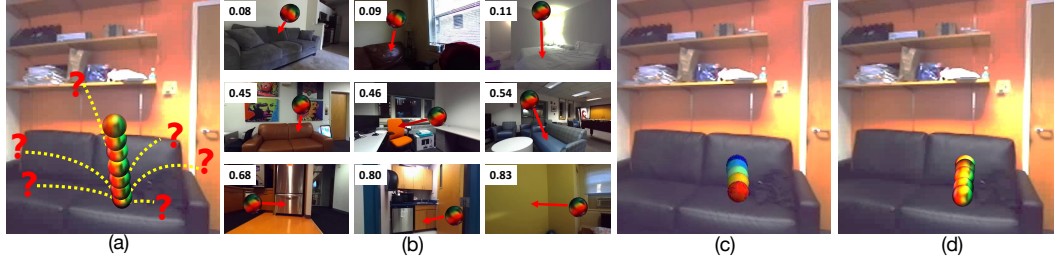

Figure 1: **Goal.** (a) We seek to predict what will happen after a ball bounces off an everyday surface. (b) We introduce a large *Bounce Dataset* of videos depicting real-world bounces in a variety of scenes. We show (hand-crafted) estimates of coefficient of restitution in the top-left corners (higher values indicate hard surfaces). (c) Output of our *Bounce and Learn* model that predicts the trajectory of the object after the bounce. (d) Observed ground truth post-bounce trajectory.

cess, which is challenging and expensive to observe and simulate, especially for softer materials (Belytschko et al., 2013; Marsden & Hughes, 2012). While there are many fast soft-body simulators (e.g., FlexSim (Nordgren, 2003)), none are physically accurate (Chen et al., 2017). Furthermore, these simulators do not allow estimation of parameters of observed real-world scenes. Moreover, current computer vision techniques do not accurately capture high-speed colliding trajectories in the wild. Inaccurate trajectory capture means we are far from certain of our trajectories – we have point clouds that are not a good fit to any exact trajectory curve but instead could be explained by any number of nearby trajectories. This means that inferring underlying collision normals and CORs cannot be done by a simple process of inverting a deterministic Newtonian physics model. Indeed, as we show in Section 4, fitting a single trajectory curve and learning with Newtonian physics leads to poor results. These results are explained in part by noting that collision dynamics and the codes that simulate them are particularly sensitive to variations and uncertainties.

To address these challenges, we seek to directly learn collision-response models of deformable surfaces from observed real-world interactions and bootstrapped by only a set of simple, inexpensive rigid-body simulation examples. We propose *Bounce and Learn*, a model that learns end-to-end to predict post-bounce trajectories and infers effective physical parameters starting from sensor inputs. Our model comprises a Physics Inference Module (PIM) and a Visual Inference Module (VIM). VIM learns to infer physical parameters for locations in a scene given a single still image, while PIM learns to model the physical interaction for the prediction task given physical parameters and an observed pre-collision 3D trajectory. We show that our model can account for non-rigid surfaces that deform during collision and, compared to inverting a parametric physics model using hand-designed features, better handles uncertainty in the captured trajectories due to end-to-end learning. Moreover, our model can be trained in batch mode over a training set or can learn to adapt in an online fashion by incorporating multiple observed bounces in a given scene. Our effort is a step towards a learnable, efficient *real-world* physics simulator trained by observing real-world interactions.

To train our model, we introduce a large-scale *Bounce Dataset* of 5K bouncing trajectories of a probe with different surfaces of varying shape and material in everyday scenes including homes and offices. For our study, we use a spherical ball made of foam for our probe as it provides a rich range of interactions with everyday objects and its symmetry allows us to better track and model the physical properties of the complex objects it collides with. As collision events are transient (we observe contact occurring over $1/50^{\text{th}}$ of a second), we have collected our dataset using a high-framerate stereo camera. Our dataset is the largest of its kind and goes well beyond simpler setups involving a handful of interacting objects (Bhat et al., 2002; Brubaker et al., 2009; Kyriazis et al., 2011; Monszpart et al., 2016). Note that prior datasets involving human interaction in sports (Bettadapura et al., 2016) require high-level reasoning without much diversity in collision surfaces.

**Contributions.** Our work demonstrates that an agent can learn to predict physical properties of surfaces in daily scenes and is the first to explore this across a large variety of different real-world surfaces, such as sofas, beds, and tables. Our contributions are twofold: (1) we propose a model that is trained end-to-end for both predicting post-bounce trajectories given an observed, noisy 3D point cloud of a pre-bounce trajectory in a scene, and for inferring physical properties (COR and collision normal) given a single still image; and (2) we build a large-scale dataset of real-world bounces in a variety of everyday scenes. We evaluate our model on our collected dataset and show that it out-

performs baselines, including trajectory fitting with Newtonian physics, in predicting post-bounce trajectories and inferring physical properties of a scene.

## 2 RELATED WORK

Our goal is related to work that captures and models physical interactions of objects in scenes. While prior work addresses various aspects of our overall goal, none gets at all aspects we seek to capture.

**Simulation-only approaches.** There have been a number of simulation-only approaches to learning or modeling object interactions and ("intuitive") physics. Examples include learning predictive models for a set of synthetic objects, like billiards (Fragkiadaki et al., 2016), and general N-body interaction problems (Battaglia et al., 2016; Chang et al., 2017; Ehrhardt et al., 2017a;b; Watters et al., 2017), and learning bounce maps (Wang et al., 2017). However, most of these approaches operate over simple scenes consisting of a single object or parametric objects. Graphics-based richer 3D environments like AI2-THOR (Zhu et al., 2017a;b) have mainly been explored for navigation and planning tasks.

**Visual prediction in toy worlds.** There are approaches that predict physical interactions in real-world imagery by incorporating a simulator during inference (Wu et al., 2015) or by learning from videos depicting real scenes (Wu et al., 2016) or simulated sequences (Lerer et al., 2016). However, these approaches model simple objects, such as blocks, balls, and ramps, and again make simplifying assumptions regarding COR, mass and friction. On the other hand, in our approach we exploit physical interactions to estimate physical properties of everyday objects in real-world scenes.

**Visual prediction in real-world scenes.** There are approaches that seek to estimate geometric and physical properties in everyday scenes using visual appearance. Examples include estimating the geometric layout of a scene (e.g., depth and surface normals from RGB-D data) (Bansal et al., 2016; Eigen & Fergus, 2015; Wang et al., 2015), material, texture, and reflectances (Bell et al., 2013), and qualitative densities of objects (Gupta et al., 2010). Instead of estimating physical properties, some approaches make visual predictions by leveraging hand-aligned Newtonian scenarios to images/videos (Mottaghi et al., 2016a), using simulated physical models fitted to the RGB-D data (Mottaghi et al., 2016b), or learning to synthesize future video frames (Chao et al., 2017; Vondrick & Torralba, 2017; Walker et al., 2016; Xue et al., 2016). A recent approach predicts sounds of everyday objects from simulations of 3D object models (Zhang et al., 2017).

**Real-world interaction and capture.** Our work is inspired by models of physical properties in the real world via interaction. Examples include parametric models of a human to model forces or affordances (Brubaker & Fleet, 2008; Brubaker et al., 2009; 2010; Zhu et al., 2015; 2016), grasping objects (Mann et al., 1997), multi-view video sequences of a ball on a ramp (Kyriazis et al., 2011), video sequences of known objects in free flight (Bhat et al., 2002), and video sequences depicting collision events between pairs of known real-world objects (Monszpart et al., 2016). We seek to generalize beyond pre-specified objects to unknown, everyday objects with complex geometries and physical properties in real-world scenes. More closely related to our work are approaches that repeatedly interact with real-world environments, such as hitting objects to learn audio-visual representations (Owens et al., 2016), crashing a drone to learn navigation (Gandhi et al., 2017), and repeated pokes and grasps of a robotic arm to learn visual representations for object manipulation (Agrawal et al., 2016; Levine et al., 2016; Pinto & Gupta, 2016; Pinto et al., 2016). Our goal is to scale learning and reasoning of dynamics starting with interactions via object bounces in everyday scenes.

## 3 BOUNCE AND LEARN MODEL AND BOUNCE DATASET

This section introduces our Bounce and Learn model and Bounce Dataset. Please see Appendix A for more details on the underlying physics governing bounces. Our overall model is shown in Figure 2 (left) and consists of a Physics Inference module (PIM) and Visual Inference Module (VIM). Having separate physics and visual modules allows for pre-training PIM using simulation data and joint training using real-world data. We describe each module in the following subsections.

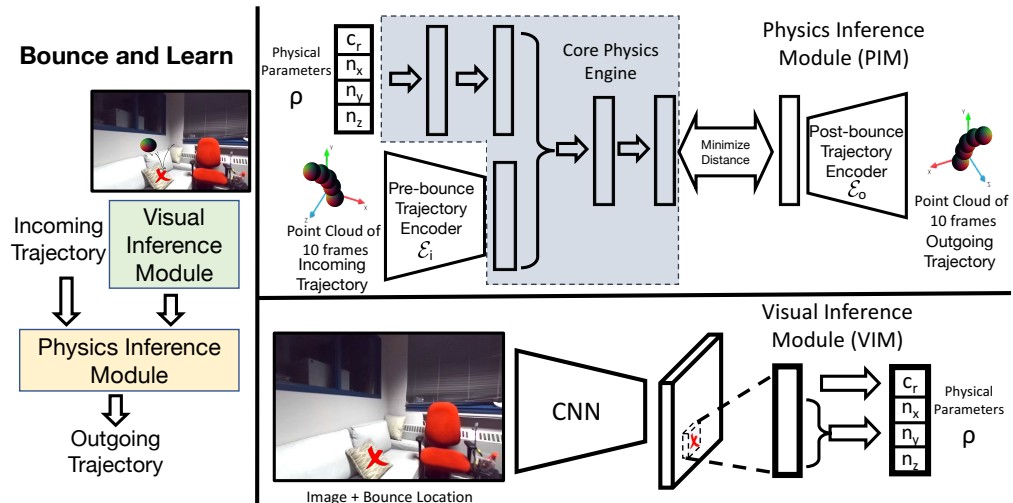

Figure 2: **System overview.** Our model (left) consists of a Physics Inference Module (top-right) and a Visual Inference Module (bottom-right). See text for more details.

## 3.1    PHYSICS INFERENCE MODULE (PIM)

PIM is shown in Figure 2 (top-right). Given a ball's incoming 3D trajectory and physical parameters of the bounce surface, the goal of PIM is to predict the outgoing 3D trajectory of the ball after bouncing off the surface. We assume the ball's trajectory is a sequence of point clouds given by the stereo camera. Let $\mathcal{T}_i$ and $\mathcal{T}_o$ be pre- and post-bounce point cloud trajectories, respectively, and $\rho$ be the physical parameters of the probed collision surface – effective collision normal and coefficient of restitution (COR). For a non-rigid collision, the input $\rho$ represents the values of effective physical parameters that lead to the aggregated effect of the impact process. We seek to learn the mapping $\mathcal{T}_o = \mathcal{F}(\mathcal{T}_i, \rho)$.

One challenge is how to represent the trajectory. While we could try to fit a static, reconstructed 3D model of the ball to the trajectory and track the centers, we found that it required manual parameter tuning. Moreover, such an approach is not easily extendable to other objects, particularly if the object undergoes deformation. We instead seek to represent the trajectory directly from sensor inputs by jointly learning an embedding in addition to predicting the post-bounce trajectory. Let $t_i$ and $t_o$ be the encoded versions of the trajectories with embedding functions $\mathcal{E}_i$ and $\mathcal{E}_o$, e.g., $t_i = \mathcal{E}_i(\mathcal{T}_i)$. Since PIM uses embedded trajectories, we can write mapping $\mathcal{F}$ as the composition,

$$\mathcal{T}_o = \mathcal{F}(\mathcal{T}_i, \rho) = \mathcal{E}_o^{-1}\left(f(\mathcal{E}_i(T_i), \rho)\right), \tag{1}$$

where core physics engine $f$ maps encoded pre-bounce trajectories $t_i$ and $\rho$ to predicted encoded post-bounce trajectories $t_p = f(t_i, \rho)$ and $\mathcal{E}_o^{-1}$ decodes $t_p$ to predict final $\mathcal{T}_o$.

**Core physics engine $f$.** We seek an architecture that is flexible and has the ability to model complex interactions, such as deformation. We use two FC layers to encode the physical parameters $\rho$ as a vector (experimentally observed to converge faster than using $\rho$ directly), which is concatenated with input encoded trajectory $t_i$ and followed by two FC layers to estimate the encoded predicted trajectory. See Appendix D for details. We observe in our experiments that core physics engine $f$, when trained on real-world data, can model interactions more complex than rigid-body collisions.

**Trajectory encoder $\mathcal{E}$.** We encode trajectories via an architecture inspired by PointNet (Qi et al., 2016). Our encoder takes as input a $T \times N \times 3$ array containing a lexicographically sorted list of 3D points, where $T$ is the number of time steps in the trajectory and $N$ is the number of 3D points at each time step. A 128-dimensional vector is generated for each time step, which are concatenated and processed by two fully connected (FC) layers, followed by $L_2$ normalization resulting in a 64-dimensional vector. See Appendix C for more details.

**Trajectory decoder $\mathcal{E}^{-1}$.** While an encoded trajectory can be decoded by a deconvolution network predicting an output point-cloud trajectory, learning a point-cloud decoder in our setting is a non-

trivial task (Fan et al., 2017); we leave this to future work. Instead, we use a non-parametric decoder. Specifically, we build a database of 10K simulated post-collision trajectories $\mathcal{T}_o$ and their encodings $t_o = \mathcal{E}_o(\mathcal{T}_o)$. We use predicted encoded trajectory $t_p$ as query and find the nearest $t_o$ to estimate $\mathcal{T}_o$.

**Pre-training PIM.** For pre-training, we need triplets $(\mathcal{T}_i, \mathcal{T}_o, \rho)$, which are easily and abundantly available from simulators (c.f., Section 3.3) that generate data under rigid body assumptions using physical parameters $\rho$. For the training loss, we minimize the distance between the encodings of the post-collision trajectory $t_o$ and the predicted post-collision trajectory $t_p$. We also use negative encoded trajectories $\{n_j\}$ to make the loss contrastive and prevent collapse of the encoding. We found that additional supervision for the trajectory encoders improves the performance of the model. We achieve this by a reconstruction network $\mathcal{P}(t_i, t_o)$ which explicitly ensures that the physical parameter vector $\rho$ can be recovered given the ground truth encoded pre- and post-bounce trajectories $t_i$ and $t_o$. We use a 2-layered neural network for $\mathcal{P}$; note that this part of the model is not shown in Figure 2 since it is only used in training and not inference. We optimize triplet and squared-$L_2$ losses for each triplet $(t_p, t_o, \{n_j\})$ corresponding to $t_i$,

$$\mathcal{L}_{\text{PIM}} = \max\left(d(t_p, t_o) - d(t_p, n_j) + m, 0\right) + ||\rho - \mathcal{P}(t_i, t_o)||_2^2, \tag{2}$$

where $d$ is cosine distance and $m$ is a scalar parameter for the margin.

**Inference.** The PIM could be potentially used for two tasks: (a) predicting post-bounce trajectories given physical parameters $\rho$ and pre-bounce trajectory $\mathcal{T}_i$; (b) estimating physical parameters $\rho$ given pre-bounce trajectory $\mathcal{T}_i$ and post-bounce trajectory $\mathcal{T}_o$. The first form of prediction, which is also used in our experiments, is straightforward: we estimate the encoded predicted trajectory $t_p$ and then use the non-parametric decoding described above. For the second task, a grid search or optimization based strategy could be used to search the range of possible physical parameters $\rho$ such that the encoded predicted trajectory $t_p$ is closest to the encoding of post-collision trajectory $t_o$.

## 3.2 Visual Inference Module (VIM)

Predicting the outcome of a bounce in a real-world scene requires knowledge of the physical parameters $\rho$ at the location of the bounce. While PIM allows us to model physical interactions, it does not provide a means to account for knowledge of the visual scene. We would like to integrate a module that can reason over visual data. To this end, we propose a Visual Inference Module (VIM) that is designed to infer the physical parameters of a scene from the visual input. We demonstrate experimentally that VIM can generalize to novel scenes for inferring physical properties and also predicting post-bounce trajectories when used in conjunction with a pretrained PIM. Moreover, in scenarios where multiple interactions with a scene is possible, we show that VIM and PIM can be jointly used to update predictions about the scene by observing the multiple-bounce events. This scenario is important since visual cues only provide limited evidence about the underlying physical properties of surfaces, e.g., two sofas that look the same visually could have different physical properties.

The proposed VIM, shown in Figure 2 (bottom-right), is a convolutional neural network (CNN) that takes as input the image of a scene and outputs the physical parameters for each location in the scene. We represent this by the function $\mathcal{V}$, which is an AlexNet architecture (Krizhevsky et al., 2012) up to the $5^{\text{th}}$ convolution layer, followed by $3 \times 3$ and $1 \times 1$ convolution layers. Each location in the output map $\mathcal{V}(\mathcal{I})$ for an image $\mathcal{I}$ contains a four-dimensional vector corresponding to the coefficient of restitution and collision normal (normalized to unit length).

**Training.** Training VIM alone is not directly possible since the ground truth for the output physical properties cannot be easily collected. These physical properties are also closely tied to the assumed physics model. Therefore, we use PIM to train VIM. For each bounce trajectory, given the impact location $(x, y)$ in the image, the physical properties can be estimated using VIM by indexing $\mathcal{V}(\mathcal{I})$ to extract the corresponding output feature. We refer to this as $\rho_{x,y}$. PIM can use the estimated $\rho_{x,y}$ along with the encoding of the pre-collision trajectory $t_i$ to predict the encoding of the post-collision trajectory. Our loss is the sum of cosine distance between the predicted and ground truth post-collision trajectory encodings $t_o$ and the squared-$L_2$ distance of $\rho_{x,y}$ from the parameters estimated with $\mathcal{P}$(described in Sec 3.1), which helps constrain the outputs to a plausible range. We also add a regularization term to ensure spatial smoothness.

$$\mathcal{L}_{\text{Joint}} = d(t_o, f(t_i, \rho_{x,y})) + ||\rho_{x,y} - \mathcal{P}(t_i, t_o)||_2^2 + \sum_{x,y} \sum_{i \in \{0,1\}} \sum_{j \in \{0,1\}} ||\rho_{x,y} - \rho_{x+i,y+j}||_2^2, \tag{3}$$

where $f$ is the core physics engine and $t_i, t_o$ are the encoded pre- and post-bounce trajectories, respectively (described in Section 3.1). The objective can be optimized by SGD to update the parameters of VIM and PIM (parameters of $\mathcal{P}$ are not updated). Note that the parameters of PIM can also be held fixed during this optimization, but we observe that training jointly performs significantly better. This demonstrates improvement over the simple rigid body physics-based pretraining. We present this ablative study in Appendix F.

**Online learning and inference (results in Appendix J).** While inference of physical properties requires physical interactions, visual cues can also be leveraged to generalize these inferences in a scene. For example, a bounce of a ball on a wall can inform our inference of physical properties of the rest of the wall. Therefore, we explore an online learning framework where our estimates of the physical parameters in a scene are updated online upon observing bounces. For every bounce trajectory $(\mathcal{T}_i, \mathcal{T}_o)$ observed at scene location $(x, y)$, we use VIM to estimate the physical parameters $\rho_{x,y}$. VIM is then updated until convergence using Equation (3). For each novel scene, we can train incrementally by interacting with the scene and updating the previously learned model. Such a setting holds significance in robotics where an agent can actively interact with an environment to make inferences about physical properties. We observe in our results that we achieve better estimates of the physical properties with an increasing number of interactions.

### 3.3 Bounce Dataset

To explore whether active physical interactions can be used to infer the physical properties of objects, we collect a large-scale dataset of a probe ball bouncing off of complex, everyday surfaces. In addition, we augment our collected real-world Bounce Dataset with a dataset of simple simulated bounces. Please see the supplemental for more details of our collected dataset.

**Real-world Bounce Dataset.** The dataset consists of 5172 stereo videos of bounces with surfaces in office and home environments. Each individual bounce video depicts the probe following a pre-collision trajectory, its impact with a surface, and its post-collision trajectory. On average each video contains 172 frames containing the ball. As shown in Figs. 1 and 5, the environments contain diverse surfaces with varying material properties. Each sample in the dataset consists of the RGB frames, depth maps, point clouds for each frame, and estimated surface normal maps. Since we are interested in the trajectory of the ball, we first perform background subtraction (Stauffer & Grimson, 1999) on the frames to localize the ball followed by RANSAC fitting (Fischler & Bolles, 1981) of a sphere on the point cloud corresponding to the foreground mask. This helps reject any outlier foreground points and collect a point cloud approximately corresponding to the ball in each frame. Note that in each frame the point cloud only depicts one viewpoint of the ball.

**Simulation data.** We bootstrap learning by augmenting our captured real-world data with simulation data. We simulate a set of sphere-to-plane collisions with the PyBullet Physics Engine (Coumans & Bai, 2016–2017). To match our captured frame rate, we set simulation time steps at 0.01 seconds. We initialize sphere locations and linear velocities randomly and set angular velocities and friction coefficients to zero. Collision surfaces are oriented randomly and COR values are sampled uniformly in the feasible range [0,1]. Each simulation returns pre- and post-bounce trajectories of the sphere for the sampled orientation and COR. To make the synthetic data consistent with our captured dataset, we create point clouds per simulation by picking a viewpoint and sampling only visible points on the sphere at each time step.

## 4 Evaluation

### 4.1 Visual Forward Prediction

For the forward-prediction task, we split our dataset into training, validation, and test sets containing 4503, 196, and 473 trajectories, respectively. There are no common scenes across these sets. We found including a mix of 3:1 synthetic-to-real data in the mini-batches achieves the best balance between prediction accuracy and interpretable physical parameter inference. Qualitative prediction results are shown in Figure 3. Notice that we effectively predict the post-bounce trajectory when the ball bounces off a variety of different surfaces. To quantitatively evaluate the model's predictions, we train on the training and validation sets and test on the test set. We report the $L_2$ distance in world

| Models | Dist | Dist Est Normal | Dist Est COR | Dist Est Normal + COR | % Normals within 30° of Est Normal | COR Median Absolute Error |
|---|---|---|---|---|---|---|
| 1. Parabola encoding | $26.3 \pm 0.5$ | $34.1 \pm 0.7$ | $26.4 \pm 0.9$ | $33.5 \pm 1.6$ | $17.52 \pm 2.22$ | $0.179 \pm 0.006$ |
| 2. Center encoding | $23.1 \pm 0.9$ | $21.2 \pm 0.3$ | $23.2 \pm 0.7$ | $21.2 \pm 0.2$ | $23.41 \pm 2.02$ | $0.178 \pm 0.013$ |
| 3. Pretrain. VIM + IN | $40.1 \pm 6.0$ | $34.1 \pm 5.4$ | $41.0 \pm 6.3$ | $34.4 \pm 5.5$ | - | - |
| 4. Ours | $21.3 \pm 0.9$ | $21.2 \pm 0.8$ | $21.4 \pm 0.7$ | $20.6 \pm 0.6$ | $24.08 \pm 3.82$ | $0.168 \pm 0.018$ |
| 5. Ours + Est. normals | $22.7 \pm 1.0$ | $18.7 \pm 0.9$ | $22.7 \pm 0.7$ | $18.4 \pm 0.9$ | $50.14 \pm 1.26$ | $0.159 \pm 0.016$ |

Table 1: **Forward prediction, collision normal estimation and COR estimation (test set).** We evaluate our model and compare to baselines on the task of forward prediction, collision normal and COR estimation. We report median distance in centimeters to observed post-bounce trajectories for each experimental setting.

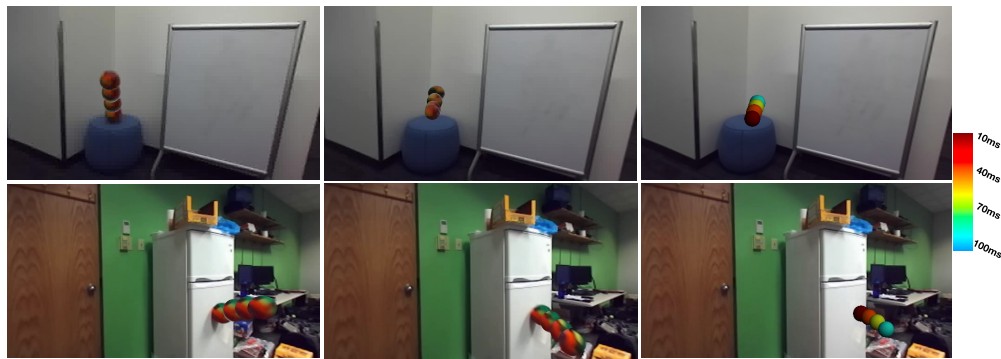

Figure 3: **Predicted post-bounce trajectories in novel scenes.** (left) Input pre-bounce trajectories. (center) Observed post-bounce trajectories. (right) Our predicted post-bounce trajectories. Additional results in Appendix.

coordinates between the predicted ball's center post-bounce and the observed ball's center extracted from the depth data at time step 0.1 seconds post-bounce (Table 1 column "Dist").

**Baselines.** The point cloud trajectory encoders in PIM are a generic solution to encoding raw trajectory data. Such a model allows applying the same model to various probe objects. An alternative model that provides less flexibility is replacing the PointNet encoder with a "**center encoding**" model, which involves extracting the center-of-mass 3D points obtained by background subtraction and extracting the mean point of the sphere point cloud at each frame for 10 frames before and after the collision. We encode the 30-D vector of $(x, y, z)$ coordinates using a two-layer neural network and train PIM with this encoding instead of the PointNet model. As a second baseline (**parabola encoding**), we fit a parabola to the set of centers and use that as the representation for the trajectory. We use the parameters of a least squares-fitted parabola as input to a two-layer neural network that outputs the encoded representation of the trajectory. We also experimented with Interaction Networks (IN) (Battaglia et al., 2016) as an alternative for the PIM. We use the output physical parameters of the VIM from our best model to predict post-bounce trajectories using IN (**Pretrain. VIM + IN**). More details about this model and direct comparison to PIM are provided in Appendix L. Note that the IN model is not trained jointly with the VIM.

**Discussion and model ablations.** Notice that our Bounce and Learn model out-performs the presented baselines (Table 1, top), likely due to the errors in hand-crafted estimates of the ball center in noisy stereo depth data. We report ablations of different training strategies for our model in Appendix F. We also quantify the effect of the spatial regularizer in Appendix G.

**Leveraging sensor-estimated collision normals and COR.** We study how our model can benefit if the collision normal or COR is known. As collision normals are loosely tied to the surface normals, we train our best model (row 4) with the sensor-estimated normals by adding an additional cosine-distance loss between the VIM-predicted collision normals and the sensor normals (row 5). Furthermore, we evaluate the different models when the sensor-estimated normals are available at test time (col. "Dist Est Normal"). Notice how most of the models benefit from this information.

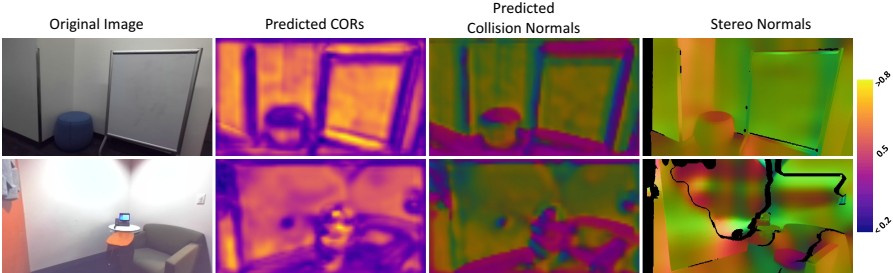

Figure 4: **Inferred COR and collision normals.** Given a single still image (first col.), we show inferred COR (second col., warmer colors indicate higher COR values), predicted collision normals (third col., colors indicate normal direction) from VIM and stereo camera surface normals (last col.).

We also investigate whether knowing a sensor-estimated COR at test time is beneficial. The COR can be estimated from pre- and post-collision velocities under the assumption of a rigid-body model. We estimate these velocities from sensor data of the ball in the frames 0.05 seconds before and after impact. Note that these are noisy estimates due to the complexity of the collisions in our dataset and errors of depth estimation from the stereo camera. We evaluate prediction accuracy when the models have access to the sensor-estimated COR during testing (col. "Dist Est COR"). We observe that for the best model, sensor-estimated COR combined with sensor normals (col. "Dist Est Normal+COR") improves accuracy over sensor-estimated COR or normals alone.

## 4.2 INFERRING PHYSICAL PROPERTIES

We show qualitative results of inferred COR values and collision normals from VIM in Figure 4 (more in Appendix I). We also compare our collision normal estimates to the surface normals estimated from the stereo camera. Notice that the soft seats have lower COR values while the walls have higher values, and the normals align with the major scene surfaces.

In Table 1, we evaluate the percentage of VIM's collision normal predictions which align within $30°$ of the sensor-estimated surface normal - the standard evaluation criterion for the NYUv2 surface normal estimation task (Silberman et al., 2012). We observe that training the trajectory encoder (row 4) leads to a model that is not constrained to predict interpretable effective physical parameters (since the input domain of $\mathcal{P}$ changes). Adding sensor-estimated normals during training improves normal prediction accuracy while maintaining good forward-prediction accuracy (row 5).

Finally, we evaluate the inferred COR from VIM. As noted in Section 4.1, the estimated COR from the stereo camera is noisy, but is the best available ground-truth. For our evaluation, we compare our inferred COR to the estimated COR from the stereo camera using median absolute error in Table 1 (last column). Notice how our best model (row 4) out-performs the baseline and ablation models.

## 5 DISCUSSION

We have introduced a new large-scale dataset of real-world bounces and have demonstrated the ability to predict post-bounce trajectories and infer physical properties of the bounces for a variety of everyday surfaces via our Bounce and Learn model. The collection of our Bounce Dataset facilitates studying physical properties not addressed by our model and future applications. Example properties include friction, object spin, surface deformation, and stochastic surfaces (e.g., corners, pebbled surface). Detection of collisions robustly and performing rollouts of predictions can be interesting directions towards practical applications. Future applications also include the ability to transfer the recovered physical properties to 3D shapes and simulating 3D scenes with real-world or cartoon physics.

## 6 ACKNOWLEDGEMENTS

This research is partly sponsored by ONR MURI N000141612007 and the ARO under Grant Number W911NF-18-1-0019. Abhinav Gupta was supported in part by Okawa Foundation.

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

## APPENDIX A    PRELIMINARIES

We review some of the fundamentals and challenges in modeling restitution during collision. Physics has long adopted simple algebraic collision laws that map a pre-collision velocity $v^-$, along with a *collision normal* $n$, to a post-collision velocity $v^+$ (Chatterjee & Ruina, 1998). Among these, perhaps the most widely applied collision law is the *coefficient of restitution* $COR = \frac{<n,v^+>}{<n,v^->} \in [0,1]$. Here $COR = 0$ gives purely inelastic impact (no rebound) and $COR = 1$ is a fully elastic impact that dissipates no energy. Note that the notion of a collision normal idealizes the impact process with a well defined normal direction. As soft objects collide (e.g., leather sofa), surface normals vary throughout the process; the collision normal $n$ encodes the averaged effect of a range of normal directions experienced throughout an impact process.

However, when it comes to physically modeling the real world, it is important to note that: (a) there is no single, physically correct COR for a material; (b) nor is there a valid geometric surface normal that encodes the collision normal behavior for a pair of colliding objects (Stewart, 2011). At best a COR value is valid only locally for a unique orientation between a pair of colliding geometries (Stoianovici & Hurmuzlu, 1996). For example, two identical rods dropped with vertical and horizontal orientations to the ground will give drastically different rebounds. If, however, a sphere with uniform material is one of the objects in a collision pair, symmetry ensures that COR will effectively vary only across the surface of the second, more complex collision surface. We leverage this in our capture setup by adopting a spherical probe object to extract a map of varying COR across complex collision surfaces in everyday environments. Similarly, the notion of a collision normal idealizes the impact process with a well defined normal direction. As soft objects collide (e.g., leather sofa), surface normals vary throughout the process. Thus a collision normal, $n$, encodes the averaged effect of a range of normal directions experienced throughout an impact process. We seek COR and collision normals that effectively encode the post-impact response.

While we could hypothetically attempt to identify materials and use material fact sheets as look-up tables to seek physical properties like COR, such information would be of limited value in any practical setting. Recall that recorded COR values are generally valid for a limited range of geometries – essentially just for perfectly rigid, sphere-to-half-plane collisions. The real world violates these assumptions almost at every turn. COR values and collision normals vary for every contact configuration and location on the object (Wang et al., 2017). There is never a single COR value characterizing collision responses against real-world objects and hence there is no database (and thus ground-truth) for materials and their COR.

## APPENDIX B    BOUNCE DATASET

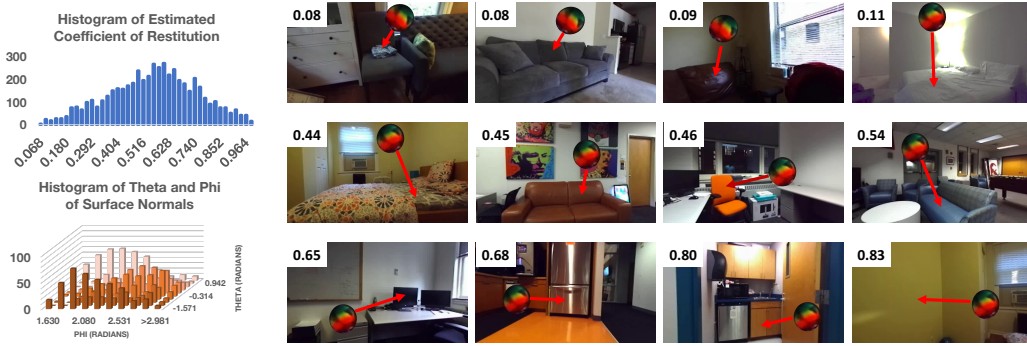

Figure 5: **Our collected Bounce Dataset of 5K real-world bounces.** Our dataset spans a variety of real-world everyday scenes. In each image we mark the location of the observed bounce and show the estimated coefficient of restitution at the top-left. Larger values correspond to harder surfaces such as floors and countertops.

We investigate whether we can use active physical interactions to infer the physical properties of objects. We explore this with a probe ball bounced off of complex, everyday surfaces to infer the

physical properties of these surfaces. Each individual bounce consists of the probe following a pre-collision trajectory, its impact with a surface, and its post-collision trajectory. In addition, we augment our collected real-world bounce dataset with a dataset of simple simulated bounces.

**Probe object.** As our focus is on probing the physical and geometric properties of complex surfaces in a scene, we apply a simple, spherical foam ball (radius ∼7 cm.) as our probe object for all our captured bounce sessions. A ball's post-collision trajectory is largely determined by the properties of the collision surface and the velocity with which the ball hits this surface. Symmetry of the ball eliminates variations in the geometry of the impact location on the probe and so its effect upon the resulting bounce.

**Capture setup.** Our analysis of bounces requires knowledge of both the pre-bounce and post-bounce trajectories of the probe object in 3D. Hence, we use the Stereolabs ZED stereo camera to capture videos at 100fps VGA resolution. We observed that the impact of the probe with a surface often takes around 1/50th of a second. The high framerate of our camera ensures that frames depicting bounces are captured. For each bounce video, we ensure that the camera is stationary to facilitate accurate reconstruction of the trajectory.

The dataset consists of 5172 stereo videos of bounces with surfaces in office and home environments. On average each video contains 172 frames containing the ball. As shown in Figure 5, these environments capture diverse surfaces with varying material properties. Each sample in the dataset consists of the RGB frames, depth maps, point clouds for each frame, and estimated surface normal maps. Since we are interested in the trajectory of the ball, we first perform background subtraction Stauffer & Grimson (1999) on the frames to localize the ball followed by RANSAC fitting Fischler & Bolles (1981) of a sphere on the point cloud corresponding to the foreground mask. This helps us reject any outlier foreground points and collect a point cloud corresponding to the ball in each frame. Note that in each frame only one viewpoint of the ball is visible. Hence, the point cloud for each frame contains a partial sphere.

**Simulation data.** We bootstrap our learning by augmenting our captured real-world data with very simple simulation data. We simulate a set of sphere-to-plane collisions with the PyBullet Physics Engine Coumans & Bai (2016–2017). To match our capture frame rate we set simulation time steps at 0.01 seconds. We initialize sphere locations and linear velocities randomly while angular velocities and friction coefficients are set to zero. Collision surfaces are likewise given a random normal orientation and COR value in the feasible range [0,1], sampling from a uniform distribution over the range of allowed values. Each simulation returns pre-bounce and post-bounce trajectories of the sphere for the sampled COR and normal. Finally, to make this synthetic data consistent with our captured dataset we create point clouds per simulation by picking a viewpoint and sampling only visible points on the sphere at each time step.

## APPENDIX C   TRAJECTORY ENCODERS

The Physics Inference Module described in Section 3.2 of the main text takes as input an encoded representation of a trajectory. Due to the significant overlap of our architecture with PointNet Qi et al. (2016), we only provide a brief description in the main text. In this section, we provide a more detailed description of the encoder model to facilitate replication of our results. Note that code for this model will be made publicly available with the final version of the paper.

Each trajectory (pre-bounce or post-bounce) in our dataset consists of point clouds of the ball at T timesteps. At each timestep, we first sample N points from the point cloud and lexicographically sort them according to their $x, y$ and $z$ coordinates (in that order of priority). We observed that random and uniform sampling achieve similar results. This sampling gives us a TxNx3 array of points for each trajectory. The lexicographic sorting provides partial spatial relationship between consecutive points in the array.

For each timestep, the corresponding $N$x3 array is processed by a sequence of convolution and ReLu layers as shown in Figure 6. This gives us a single 1x128 dimensional vector for each time step. These vectors for each timestep are concatenated to give us a 1x128T dimensional vector. This vector is processed by two fully connected layers followed by L2 normalization giving us a 256-

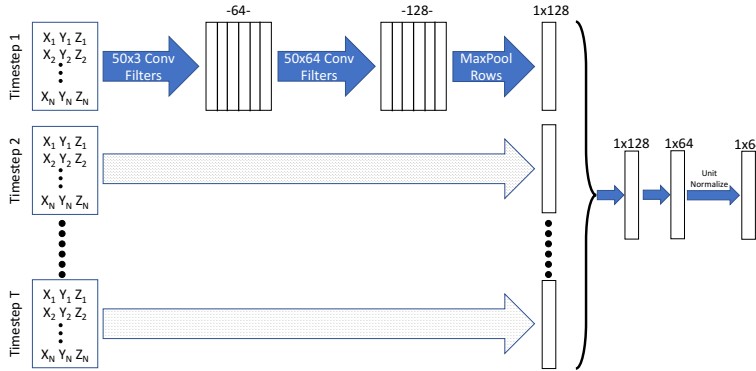

Figure 6: Our proposed encoder architecture takes as input a sequence of length $T$ of point cloud data, each containing $N$ points, and outputs a single vector.

dimensional vector. We observed that using 64-dimensions as the encoding size leads to the best results. In our implementation, we set T=10 and N=500.

## APPENDIX D  CORE PHYSICS ENGINE

The Core Physics Engine described in Section 3.2 is the component of the PIM that performs the prediction task. Given input physical parameters $\rho$ and an encoded pre-bounce trajectory $t_i$, the Core Physics Engine predicts the encoded post-bounce trajectory $t_o$. This is done by first encoding the input physical parameters in a suitable latent representation $v_\rho$ (1x32 vector in our implementation) using two fully connected layers. We observed that increasing the dimensionality of the interme-diate representation $v_p$ beyond 1x32 doesn't improve results and using sizes below 1x32 leads to a drop in performance. The encoded physical parameters $v_\rho$ are then concatenated with the encoded pre-bounce trajectory $t_i$. This concatenated vector is used to predict the post-bounce trajectory en-coding $t_o$ using two fully connected layers. This pipeline along with the sizes of the intermediate representations are shown in Figure 7.

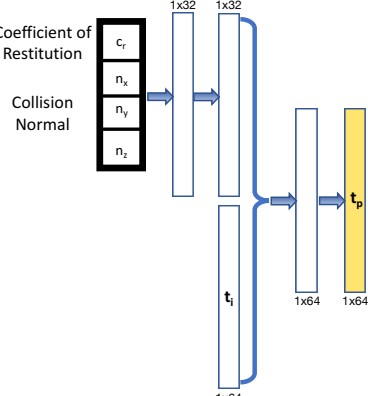

Figure 7: Our proposed Core Physics Engine takes as input the encoded pre-bounce trajectory $t_i$ along with the physical parameters of the collision surface $\rho$ to predict the encoded post-bounce trajectory $t_p$.

## APPENDIX E    TRAINING DETAILS

### E.1    PRETRAINING THE PIM

The Physics Inference Module (PIM) is pretrained using simulation data as described in Section 3.2. In order to optimize the objective function given in Equation 2 of the main text we use the Adam optimizer. We update the parameters of the PIM using a batchsize of 32, initial learning rate of 0.01 and weight decay of 0.0005. The learning rate is dropped by a factor of 10 after every 32000 iterations. The training is done for a total of 96000 iterations.

### E.2    TRAINING VIM+PIM

The Visual Inference Module (VIM) is trained jointly with the PIM using the captured data as described in Section 3.3. Similar to the PIM, the objective presented in Equation 3 of the main text is optimized using the Adam optimizer. We update the parameters of the PIM and VIM using a batchsize of 32, initial learning rate of 0.001 and weight decay of 0.0005. The learning rate is dropped by a factor of 10 after every 8000 iterations. We observe that the training converges at 24000 iterations.

## APPENDIX F    ABLATIVE STUDY: TRAINING VARIOUS COMPONENTS OF THE BOUNCE AND LEARN PIPELINE

We conducted an ablative study on training various components of the Bounce and Learn pipeline. These results are presented in Table 2. First, we consider pre-training PIM (core physics engine and trajectory encoder) with only synthetic data and holding it fixed when training VIM on real data (row 1). In other words, PIM only sees synthetic data during training and is upper bounded by the rigid-body Newtonian physics simulation. Next, we consider pre-training PIM on synthetic data and fine-tuning with VIM training on real data (row 2). Next, we consider pre-training PIM on synthetic data and fine-tuning the core physics engine (holding the trajectory encoder fixed) with VIM training on real data (row 3).

| Training Setting | Dist | Dist Est Normal | Dist Est COR | Dist Est Normal + COR | % Normals within 30° of Est Normal | COR Median Absolute Error |
|---|---|---|---|---|---|---|
| 1. Fix core and traj. enc. (synth only) | $37.2 \pm 1.5$ | $29.6 \pm 0.4$ | $38.3 \pm 0.4$ | $29.4 \pm 0.1$ | $28.72 \pm 4.87$ | $0.193 \pm 0.020$ |
| 2. Train core and traj. enc. | $20.4 \pm 0.9$ | $31.7 \pm 2.8$ | $20.4 \pm 2.5$ | $31.4 \pm 2.0$ | $13.01 \pm 1.26$ | $0.134 \pm 0.008$ |
| 3. (Ours) Train core, Fix traj. enc. | $19.9 \pm 1.8$ | $18.1 \pm 0.8$ | $20.5 \pm 1.0$ | $17.8 \pm 0.7$ | $34.98 \pm 3.43$ | $0.123 \pm 0.009$ |

Table 2:  **Bounce and Learn ablative evaluation (val set).** We evaluate our models in different training settings on the task of forward prediction, collision normal and COR estimation. We report median distance in centimeters to observed post-bounce trajectories for each experimental setting. Please see the text for details.

We observe that fine-tuning PIM on real data results in better prediction accuracy than training PIM on synthetic data alone. We also observe that our best model requires finetuning the core physics engine while keeping the trajectory encoding fixed. This suggests that the core physics engine learns effective physical parameters beyond the parameters used in rigid-body simulation pretraining. We further analyze this hypothesis in Appendix H.

## APPENDIX G    ABLATIVE STUDY OF SPATIAL REGULARIZATION

Here we analyze the effect of the spatial regularizer term presented in Equation 3. We evaluate forward prediction, normal estimation and COR estimation errors for our proposed model with and without spatial regularization ("Reg."). These results are presented in Table 3.

| Models | Dist | Dist Est Normal | Dist Est COR | Dist Est Normal + COR | % Normals within 30° of Est Normal | COR Median Absolute Error |
|---|---|---|---|---|---|---|
| 1. (Ours) Point Cloud-based | 20.5± 1.1 | 18.5± 0.6 | 20.3± 1.3 | 18.2± 0.5 | 34.14± 2.87 | 0.127± 0.008 |
| 2. (Ours) Point Cloud-based + Spatial Reg. | 19.9± 1.8 | 18.1± 0.8 | 20.5± 1.0 | 17.8± 0.7 | 34.98± 3.43 | 0.123± 0.009 |

Table 3: **Spatial Regularization Ablation (val set).** We present an ablative study of the effect of spatial regulation on the Bounce and Learn model. We report median distance in centimeters to observed post-bounce trajectories for each experimental setting. We also evaluate the predicted normals and CORs.

## APPENDIX H    ANALYZING THE EFFECT OF TRAINING PIM ON REAL DATA

In the results presented in Section 4 of the main text, we observe that jointly training the core physics engine leads to significant improvements in performance on both tasks - forward prediction and physical parameter estimation. Here we further analyze this by looking at the forward prediction errors for trajectories within different ranges of sensor-estimated COR (calculated using the hand-crafted approach described in Sec 4.1). We present these results in Figure 8.

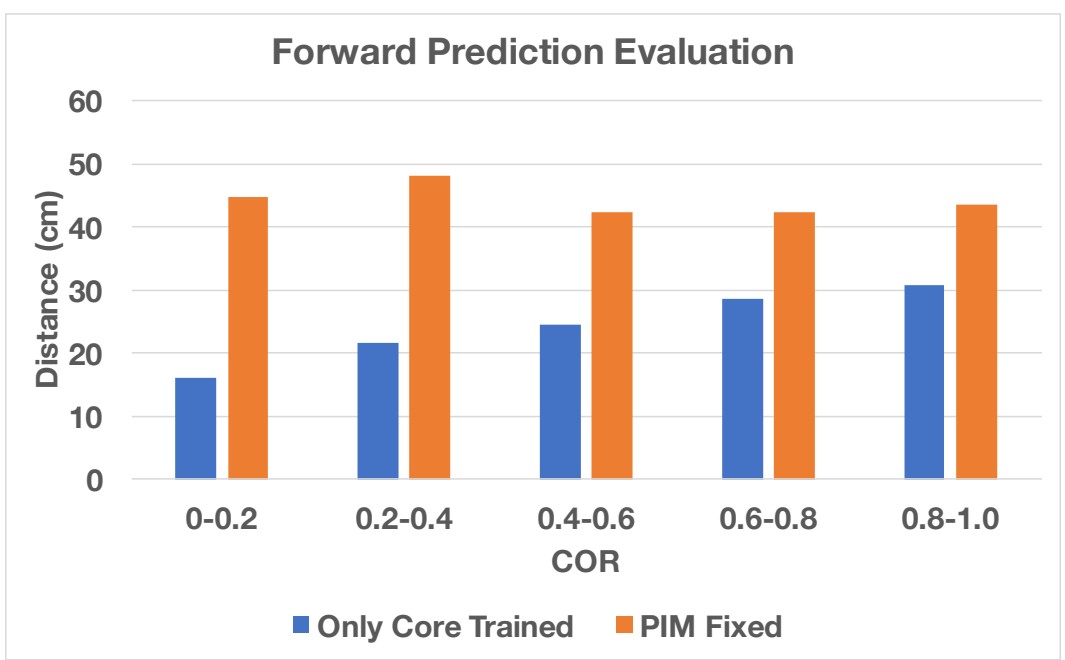

Figure 8: Forward prediction errors for trajectories belonging to different ranges of estimated COR. An interesting observation is that the improvement in performance in lower COR ranges is significantly higher.

In general, lower COR values point to collisions that are more non-rigid and higher COR value collisions are closer to rigid-body collisions. While there are exceptions to this intuition (for example, trampolines are high COR and non-rigid), such examples are rare in the Bounce dataset. We observe from the results in Figure 8 that the improvement in performance is significantly higher in the lower COR ranges compared to the higher. This supports the hypothesis that training the PIM jointly leads to modeling of non-rigid collisions better than the pretrained rigid-body simulation based PIM. Furthermore, we see that training the PIM also improves forward prediction results in all ranges of COR. It is also worth noting that the predictions are generally more sensitive to variations in the higher COR region leading since small variations in velocities could lead to larger distance errors. This explains the increasing error for the "Only Core Trained" model as the values of the sensor-estimated COR increases.

## APPENDIX I   ADDITIONAL QUALITATIVE RESULTS

We provide additional visualizations of the predictions from the VIM in Figure 9. We also provide interesting visualizations of the predicted trajectories by the Bounce and Learn model in Figure 10. Row 1 of Figure 10 shows a case where the restitution is predicted to be slightly higher. Row 3 of Figure 10 shows a case where the output seems physically plausible, but does not match the true trajectory since it fails to account for the spin of the ball.

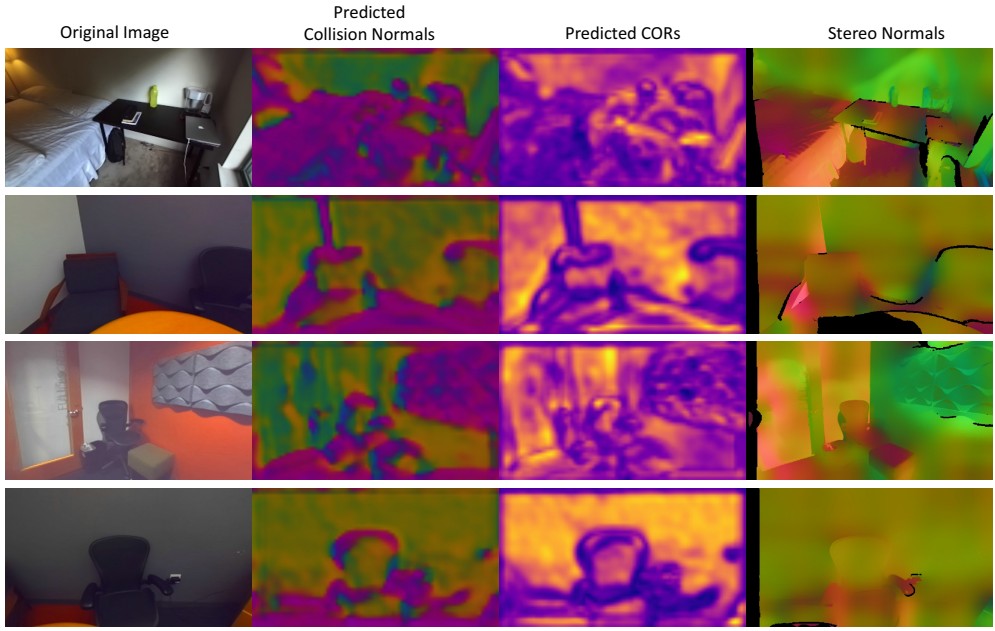

Figure 9: **Inferred COR and collision normals.** Given a single still image (first col.), we show inferred COR (second col., warmer colors indicate higher COR values), predicted collision normals (third col., colors indicate normal direction) from VIM and stereo camera surface normals (last col.).

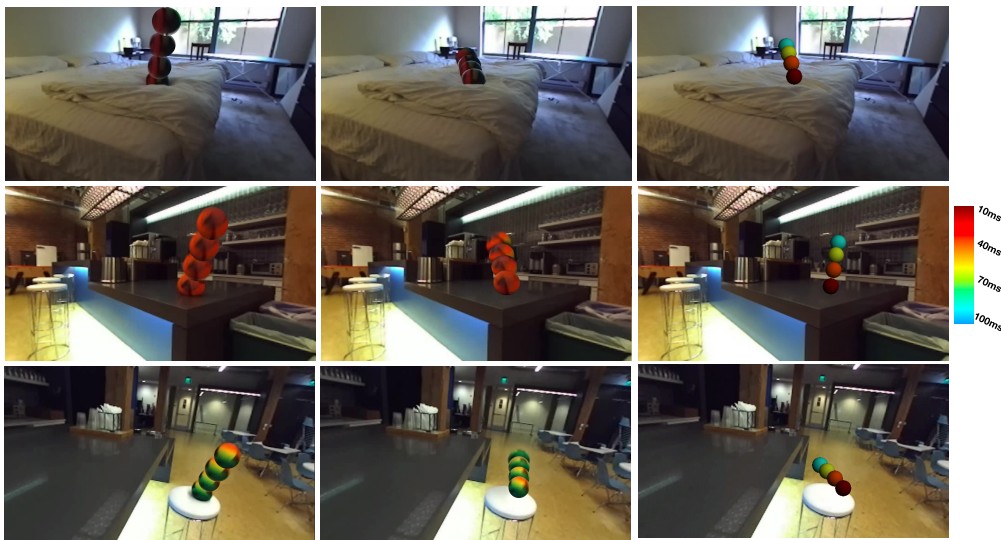

Figure 10: **Predicted post-bounce trajectories in novel scenes.** (left) Input pre-bounce trajectories. (center) Observed post-bounce trajectories. (right) Our predicted post-bounce trajectories.

## APPENDIX J   ONLINE INFERENCE FROM BOUNCES AND VISUAL CUES

In the main text, we demonstrate the efficacy of the VIM+PIM framework for inference of physical properties and predicting bounce outcomes in novel environments. However, in numerous robotics applications, an agent can usually interact with a scene to infer the physical properties. In such interactions, visual cues can also be leveraged to generalize these inferences in a scene. For example, a bounce of a ball on a wall can inform our inference of physical properties of the rest of the wall. Therefore, we explore an online learning framework, where our estimates of the physical parameters in a scene are updated online upon observing bounces.

First, we pretrain the PIM as described in Section 3.2. For every bounce trajectory $(\mathcal{T}_i, \mathcal{T}_o)$ observed at scene location $(x, y)$, we use the VIM to estimate the physical parameters $\rho_{x,y}$. The VIM is then updated until convergence using the same objective function as the VIM training from Equation 3 (also presented here).

$$\mathcal{L} = d(t_o, f(t_i, \rho_{x,y})) + ||\rho_{x,y} - \mathcal{P}(t_i, t_o)||_2^2$$

The loss is optimized incrementally using all observed bounces so far. Therefore, for each scene, we can train incrementally by interacting with the scene and updating the previously learned model. The PIM is then kept fixed during the online learning process. Fixing the PIM makes the optimization easier since we usually have access to limited number of bounce trajectories in novel environments (interactions of the agent).

We observe in our results that we achieve better estimates of the physical properties with an increasing number of interactions. In Figure 11, we visualize the intermediate predictions from the VIM after observing an increasing number of bounces. Evidently, the estimates of collision normals and coefficient of restitution for the image improves with the number of bounces. For each scene, we leave out 10 bounce trajectories for evaluation and use the rest as the online interactions. We consider 10 random shuffles of the bounce trajectories, creating different sets for evaluation and online interactions, to compute mean performance. Figure 11 shows the quantitative improvement with increasing number of bounces according to the metrics described in Section 5.2. We present the final predictions from two other scenes in Figure 12. Observe that the predictions accurately capture the lower restitution of softer objects (pillow, seat of chair). Similarly, it also captures the "hardness" of the edge of the chair and tables. Next, we demonstrate the prediction of trajectories. Qualitative examples of trajectory predictions in the online setting are shown in Figure 13. We observe that the combination of VIM and PIM can successfully predict the outcomes of most bounce events. We also present some failure cases in Figure 13.

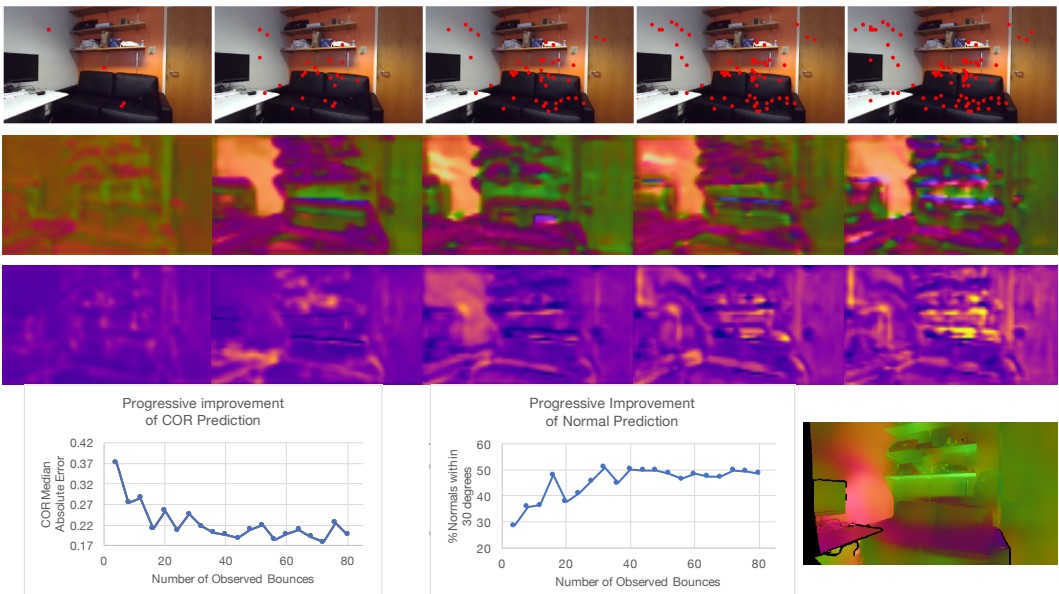

Figure 11: We learn to estimate physical parameters of surfaces in a scene by observing bounces at different locations (row 1). Our predictions of collision normals (row 2) and coefficient of restitution (row 3) improve with number of bounces. The quantitative evaluation (row 4) provides strong evidence for the efficacy of our online learning approach.(Best viewed electronically)

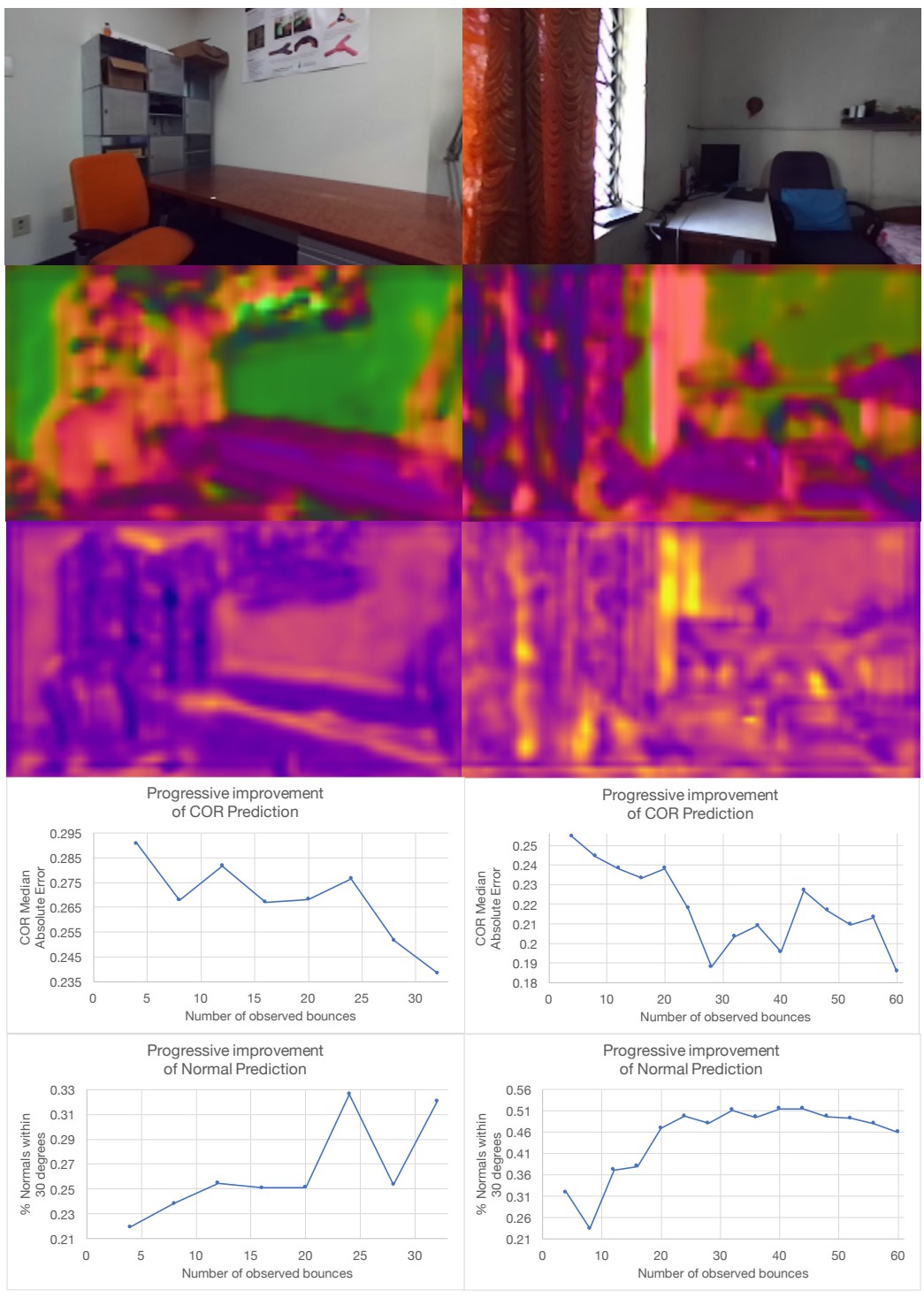

Figure 12: The final predicted COR maps shows the ability of our approach to differentiate soft (pillows, moveable lamp) and rigid objects (edges of chairs and tables). Furthermore, the quantitative evaluation of the estimated collision normals and COR shows a strong improving trend with increasing number of bounces.(Best viewed electronically)

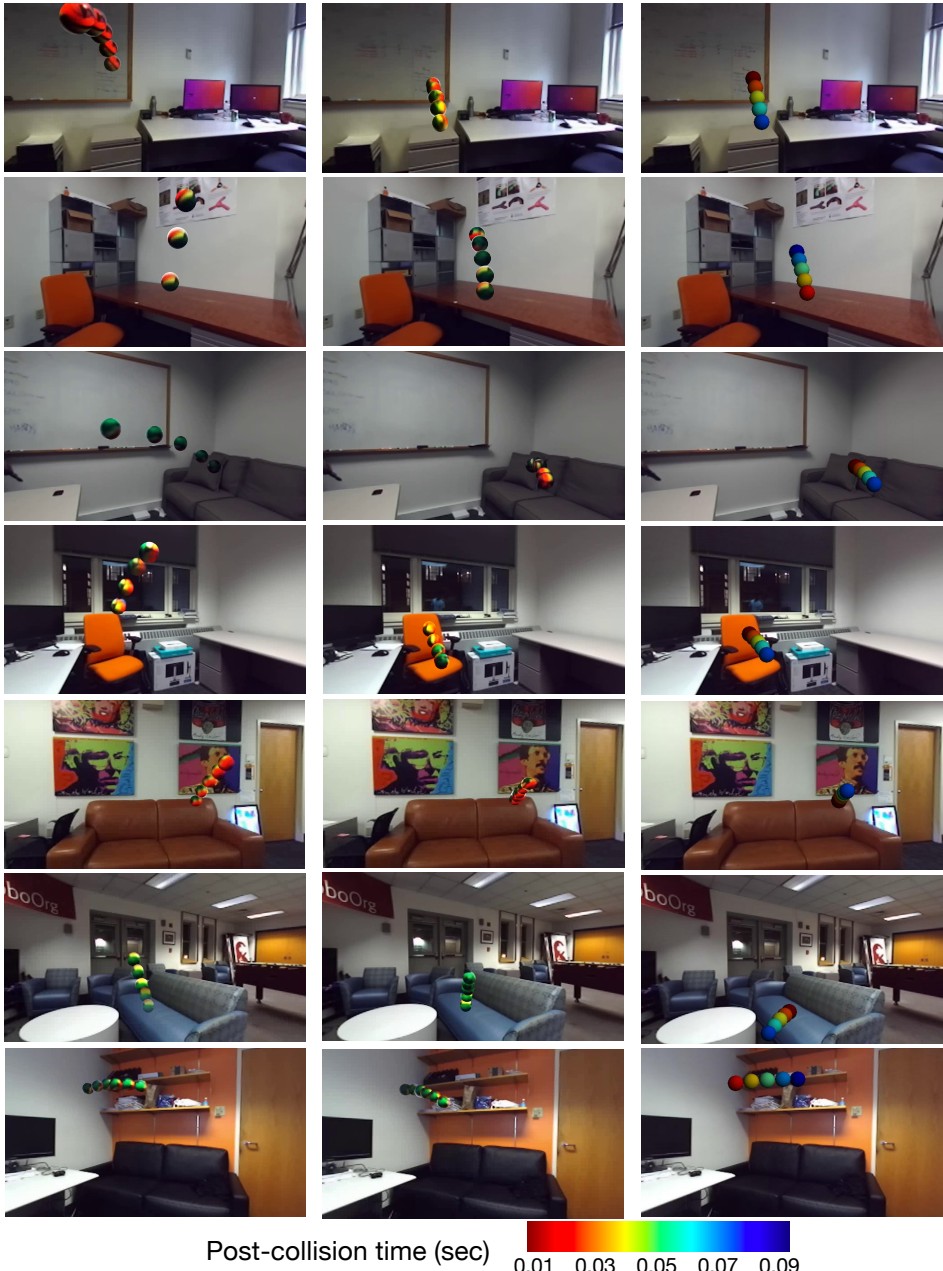

Figure 13: **Online Learning based Predicted post-bounce trajectories.** (left) Input pre-bounce trajectories. (center) Observed post-bounce trajectories. (right) Our predicted post-bounce trajectories. We correctly predict the trajectory in different scenes. Bottom two rows are example failures.

## APPENDIX K  EVALUATION IN THE ABSENCE OF IMPACT LOCATION ANNOTATIONS

| Models | Dist | Dist Est Normal | Dist Est COR | Dist Est Normal + COR | % Normals within 30° of Est Normal | COR Median Absolute Error |
|---|---|---|---|---|---|---|
| Annotated collision pts | 21.3± 0.9 | 21.2± 0.8 | 21.4± 0.7 | 20.6± 0.6 | 24.08± 3.82 | 0.168± 0.018 |
| Estimated collision pts | 21.8± 1.7 | 21.5± 0.1 | 22.0± 1.8 | 21.4± 0.5 | 24.26± 1.99 | 0.171± 0.023 |

Table 4: **Bounce and Learn with estimated collision points (test set).** We evaluate our model on the task of forward prediction, collision normal and restitution estimation under the absence of impact location annotations in the scene image.

In the experiments presented in Section 4, we use human-annotations for the impact location in the image. These annotations are used to index the output of the Visual Inference Module (VIM) as explained in Equation 3. In this section, we conduct an experiment by relaxing the need for this human annotation. The point cloud of the ball in the frame of collision provides information about it's location in the image. We leverage this information to estimate the point of collision in the image. More specifically, we project the mean point of the point cloud to the image using the camera parameters. This serves as an estimate of the point of collision. Note that this is not an accurate estimate since the point of collision could occur at the edges of the ball, which would not coincide with the projected center. However, since the output of VIM is coarse, we hypothesize that minor errors in impact location would not effect the coarse index. In Table 4, we present results for our model trained and tested using this estimate for collision location. We observe very minimal difference in performance compared to using the human annotations.

## APPENDIX L  COMPARING PIM TO INTERACTION NETWORKS (BATTAGLIA ET AL., 2016)

| Models | Dist |
|---|---|
| Center-based PIM | 10.87± (0.32) |
| IN - velocity | 49.00± (6.34) |
| IN - location history | 20.10± (1.50) |

Table 5: **Comparison to Interaction Networks.** We evaluate our Center-based PIM model and Interaction Networks (Battaglia et al., 2016) on 10000 simulated trajectories. We report median distance in centimeters to ground truth post-bounce location at t=0.1s. Please see the text for details.

Interaction Networks (IN) (Battaglia et al., 2016) model the interactions between objects in complex systems by performing inferences about the abstract properties that govern them. INs are shown to be effective for reasoning about n-body interactions, colliding rigid balls and interaction of discretized strings with rigid objects. The Physics Inference Module (PIM) proposed in this paper is aimed to address a more specific problem - collision between two non-rigid objects. However, the PIM also provides the additional benefit of performing inference over sensor inputs in the form of point clouds.

We perform a quantitative comparison of the PIM and IN models. Since the IN is not designed to deal with point cloud data, we use the Center-based PIM baseline presented in Section 4 for fair comparison. The IN model maintains a state vector of each object at each timestep. We follow the choices in (Battaglia et al., 2016) to design the state vector. In our scenario, we have two objects - the ball and the collision surface. We represent both using a 7-dimensional state vector. The surface is represented as a static object with inverse mass 0, located at (0, 0, 0) and with velocity (0, 0, 0). The ball is represented as an object with constant inverse-mass and with the location and velocity determined by the sample. The relation attribute is represented by the coefficient of restitution and the collision normal. Since our training and test simulation trajectories have added noise to imitate

sensor data, the estimates of initial velocity are also noisy. Therefore, we formulate another location history based state vector containing location of the object in the last 3 time steps. More concretely, the history-based state vector is represented as:

$$o = [m, x_{t-2}, y_{t-2}, z_{t-2}, x_{t-1}, y_{t-1}, z_{t-1}, x_t, y_t, z_t]$$

We test these models on 10000 simulated trajectories and present the results in Table 5. We observe that the IN model demonstrates a higher error at t=0.1s post-collision. We believe that this difference in error is due to the recurrent nature of IN, leading to accumulation of errors at every time step. On the other hand, since the PIM predicts by choosing the best simulated post-collision trajectory, it does not suffer from this issue. However, incorporating IN in our proposed framework could allow modelling of n-body interactions and could be addressed in future work.

