# OpenReview forum: "Bounce and Learn: Modeling Scene Dynamics with Real-World Bounces"
_ICLR.cc/2019/Conference_

### Official Review · AnonReviewer3 · 2018-10-15
**Great work; Important and interesting problem; Missing some details**

**Rating:** 8
**Confidence:** 4

**Review:**

Paper summary:
The paper proposes to predict bouncing behavior from visual data. The model has two main components: (1) Physics Interface Module, which predicts the output trajectory from a given incoming trajectory and the physical properties of the contact surface. (2) Visual Interface Module, which predicts the surface properties from a single image and the impact location. A new dataset called Bounce Dataset is proposed for this task.

Paper strengths:
- The paper tackles an interesting and important problem.
- The data has been collected in various real scenes.
- The idea of training the physics part of the network with synthetic data and later fine-tuning it with real images is interesting.
- The experiments are thorough and well-thought-out.

Paper weaknesses:
- It would be more interesting if the dataset was created using multiple types of probe objects. Currently, it is only a ball.

- It is not clear how the evaluation is performed. For instance, the length of the groundtruth and predicted trajectories might be different. How is the difference computed?

- The impact location (x,y) corresponds to multiple locations in 3D. Why not using a 3D point as input? It seems the 3D information is available for both the real and synthetic cases.

- Why is it non-trivial to use a deconvolution network for predicting the output point cloud trajectory?

- The length of the input trajectory can vary, but it seems the proposed architecture assumes a fixed-length trajectory. I am wondering how it handles a variable-length input.

- How is the bounce location encoded in VIM?

- I don't see any statistics about the objects being used for data collection. That should be added to the paper.

>>>>> Final score: The authors have addressed my concerns in the rebuttal. I believe this paper tackles an interesting problem, and the experiments are good enough since this is one of the first papers that tackle this problem. So I keep the initial score.

---

> ### Author Response · Authors · 2018-11-15
> **Author Response for AnonReviewer3**
>
> We thank the reviewer for their appreciation of our work. We address the reviewer’s concerns here:
>
> 1) “It would be more interesting if the dataset was created using multiple types of probe objects. Currently, it is only a ball.”
>         We agree that the eventual goal for research in this direction should be to generalize to multiple types of probe objects. We discuss this further in the response to the review from AnonReviewer2. (https://openreview.net/forum?id=BJxssoA5KX&noteId=ByekAQQo6Q )
>
> 2)“The length of the groundtruth and predicted trajectories might be different. How is the difference computed?”
>         The evaluation is not dependent on the length of the trajectories recorded. The distance between the predicted center and the ground-truth center is computed at timestep 10 (0.1 seconds post-bounce). All trajectories in the dataset have length greater than 10 timesteps.
>
> 3) “The impact location (x,y) corresponds to multiple locations in 3D. Why not using a 3D point as input? It seems the 3D information is available for both the real and synthetic cases.”
>         In the physics model, the 3D collision point is currently used since the point cloud is represented with collision as origin. In the VIM model, using the 3D points is similar to using a 2D (x,y) points since we eventually need to extract visual features from 2D input images.
>
> 4) “Why is it non-trivial to use a deconvolution network for predicting the output point cloud trajectory?”
>         There is very limited work on generating point clouds from embeddings. Integrating a deconvolution model would have added an additional obstacle to an already challenging problem. Furthermore, it would make localizing the errors more difficult.
>
> Some relevant literature that demonstrate the challenges of generating point clouds:
> [1] Achlioptas, Panos, et al. "Representation learning and adversarial generation of 3D point clouds." arXiv preprint arXiv:1707.02392 (2017).
> [2] Insafutdinov, Eldar, and Alexey Dosovitskiy. "Unsupervised Learning of Shape and Pose with Differentiable Point Clouds." arXiv preprint arXiv:1810.09381 (2018).
> [3] Lin, Chen-Hsuan, Chen Kong, and Simon Lucey. "Learning efficient point cloud generation for dense 3D object reconstruction." arXiv preprint arXiv:1706.07036 (2017).
> [4] Achlioptas, Panos, et al. "Learning Representations and Generative Models for 3D Point Clouds." (2018).
>
>
> 5) “The length of the input trajectory can vary, but it seems the proposed architecture assumes a fixed-length trajectory. I am wondering how it handles a variable-length input.”
>         We observed that 10 frames before and after the collision contain sufficient information. Therefore, we used these 20 frames in the proposed model. For videos where more frames are available, we use only the 10 frames before and after collision.
>
> 6) “How is the bounce location encoded in VIM?”
>         The bounce location is used to index the feature map which is the output of the VIM. We present this in Subsection 3.2 “Training” paragraph - $\rho_{x,y}$ is obtained by indexing the output $\mathcal{V}(I)$.
>
> 7) “I don't see any statistics about the objects being used for data collection. That should be added to the paper.”
>         Thank you for the suggestion. That would indeed be informative. We shall add this to the final version of the paper since this would require some additional effort to label the objects.

---

### Official Review · AnonReviewer1 · 2018-11-02
**a well evaluated solution to an interesting and challenging problem**

**Rating:** 7
**Confidence:** 4

**Review:**

This paper presents a method for inferring physical properties of the world (specifically, normals and coefficients of restitution) from both visual and dynamic information.  Objects are represented as trajectories of point clouds used under an encoder/decoder neural network architecture.  Another network is then learned to predict the post bounce trajectory representation given the prebounce trajectory representation given the surface parameters.  This is used both to predict the post bound trajectory (with a forward pass) but also to estimate the surface parameters through an optimization procedure.  This is coupled with a network which attempts to learn these properties from visual cues as well.  This model can be either pretrained and fixed or updated to account for new information about a scene.

The proposed model is trained on a newly collected dataset that includes a mixture of real sequences (with RGB, depth, surface normals, etc) and simulated sequences (additionally with physical parameters) generated with the help of a physics engine.  It is compared with a number of relevant baseline approaches and ablation models.  The results suggest that the proposed model is effective at estimating the physical properties of the scene.

Overall the paper is well written and thoroughly evaluated.  The problem is interesting and novel, the collected dataset is likely to be useful and the proposed solution to the problem is reasonable.

---

> ### Author Response · Authors · 2018-11-15
> **Author Response for AnonReviewer1**
>
> We thank the reviewer for their time and appreciation of our work.

---

### Official Review · AnonReviewer2 · 2018-11-05
**Might be Good but Difficult to Evaluate:  No Comparison to Existing Methods.**

**Rating:** 6
**Confidence:** 3

**Review:**

The authors present both a dataset of videos of a real-world foam ball bouncing and a model to learn the trajectory of the ball at collision (bounce) points in these videos.  The model is comprised of a Physics Inference Module (PIM) and a Visual Inference Module (VIM).  The PIM takes in both a vector of physical parameters (coefficient of restitution and collision normal) and a point cloud representation of the pre-bounce trajectory, and produces a point cloud representation of the post-bounce trajectory (or, rather, an encoded version of such).  The VIM takes in an image and ground-truth bounce location and produces the physical parameters of the surface at that location.

I find the paper well-written and clear.  The motivation in the introduction is persuasive and the related work section is complete.  However, the authors are introducing both a new training paradigm (to my knowledge unused in the literature) and a new model, and without any existing baselines to compare against I find it a bit difficult to understand how well the model works.

Overall, the authors’ model is somewhat complicated and not as general as it initially seems.  To justify this complication I would like to see more convincing results and benchmarking or application to more than one single dataset (e.g. non-spheres bouncing).

Here are some specific concerns:

1)  I could not find a link to an open-sourced version of the dataset(s).  Given that the authors emphasize the dataset as a main contribution of the paper, they should open-source it and make the link prominent in the main text (apologies if I somehow missed it).

2)  The authors claim in multiple places that the model is trained end-to-end, but this does not seem to be the case.  Specifically, the PIM is pre-trained on an auxiliary dataset from simulation.  The trajectory encoder also seems to be pre-trained (though I could be wrong about that, see my question below).  Furthermore, there is a bit of hand-holding:  The PIM uses ground-truth state for pre-training, and the VIM gets the ground-truth bounce location.  In light of this, the model seems a lot less general and end-to-end than implied in the abstract and introduction.

3)  No comparison to existing baselines.  I would like to see how the authors’ model compares to standard video prediction algorithms.  The authors could evaluate their model with respect to pixel loss (after ground-truth rendering) and compare to a video prediction algorithm (such as PredNet by Lotter, Kreiman, & Cox, 2016).  Given that the authors’ method uses some extra “privileged” information (as described in point 2), it should far out-perform algorithms that train only on video data, and such a result would strengthen the paper a lot.

4)  Table 1 is not a very convincing demonstration of performance.  Regardless of baselines, the table does not show confidence intervals.  I would love to see training curves with errorbars of the models on the most important metrics (e.g. Dist and COR Median Absolute Error).

I also was confused about a couple of things:

1)  How was the PointNet trajectory encoder trained?  I did not see this mentioned anywhere.  Were gradients passed through from the PIM?  Was the same network used for both the simulation and real-world data?

2)  The performance of the center-based model in Table 1 seems surprisingly low.  The center-based model should be as good at the Train core, Fix traj. enc. model, since it has access to the ball’s position.  Why is it worse?  Is the VIM at fault?  Or is the sphere-fitting sub-optimal?  How does it compare on the simulated data with ground truth physical parameters?

3)  Lastly, the color-scheme is a bit confusing.  It looks like the foam ball in the videos was rainbow-colored.  However, in the model outputs in trajectory figures time is also rainbow-colored.  This was initially a bit confusing.  Perhaps grayscale for the model outputs would be clearer.

---

> ### Author Response · Authors · 2018-11-15
> **Author Response for AnonReviewer2**
>
> We thank the reviewer for their feedback. We address the concerns of the reviewer below.
>
> 1) “The authors are introducing both a new training paradigm (to my knowledge unused in the literature) and a new model, and without any existing baselines to compare against I find it a bit difficult to understand how well the model works.”
>         We agree that due to the novelty of our training paradigm, model and data, there is a lack of existing literature/baselines to compare against. This is an unavoidable challenge we face. However, in order to better provide context for the performance of our models, we have conducted extensive quantitative and qualitative experiments and compared to relevant baselines (as also noted by other reviewers) including: (a) experiments dissecting the proposed model to localize the performance gains obtained due the PointNet trajectory encoders; (b) training the PIM on real-world data; and (c) a ground truth normals based experiment for reference. Overall, we hope that our proposed approach can also serve as a useful baseline for future work in this direction.
>
> 2) “Overall, the authors’ model is somewhat complicated and not as general as it initially seems.  To justify this complication I would like to see more convincing results and benchmarking or application to more than one single dataset (e.g. non-spheres bouncing).”
>         As previously noted by the reviewer, prior work along the lines of estimating physical parameters and learning models of physics from real-world data is extremely scarce. Therefore, there are no relevant datasets that can directly be used to benchmark our approach, which also emphasizes the need for such a dataset.
> In the nascent stages of this field, we believe that addressing the problem with a spherical probe object provides a good starting point. Non-spherical probe objects introduce additional complexity making exploration in this direction more challenging. For example, results in [a] show how much the physical properties vary across the surface of an object. The controlled setup of a spherical probe object ensures that the outcomes of bounces are dependent only on the physical properties of one object. However, we agree that non-spherical probe objects could definitely be an interesting and essential next step to pursue as future work.
>
> Specific concerns:
> 1) “A link to open source version of dataset is not available”
>         The double-blind submission of ICLR constrains the ability for us to provide the dataset publicly without revealing our identity. The data will be made publicly available with the final version of the paper.
>
> 2) “The authors claim in multiple places that the model is trained end-to-end, but this does not seem to be the case.  Specifically, the PIM is pre-trained on an auxiliary dataset from simulation.  The trajectory encoder also seems to be pre-trained (though I could be wrong about that, see my question below).  Furthermore, there is a bit of hand-holding:  The PIM uses ground-truth state for pre-training, and the VIM gets the ground-truth bounce location.  In light of this, the model seems a lot less general and end-to-end than implied in the abstract and introduction.”
>         The PIM (including the trajectory encoder) is pretrained initially using simulation data. The VIM+PIM pipeline is then finetuned in end-to-end manner on the real data. In the abstract/introduction, we refer to this end-to-end training. It is true that the PIM uses simulation parameters in the pretraining phase and the VIM uses the ground truth location to index the feature maps. However, the training is still “end-to-end”  in the conventional usage of the term, since the model is fully differentiable and the gradients for the objective in Equation 3 are computed w.r.t all the parameters of both the VIM and PIM. This is analogous to pretraining on ImageNet and finetuning with added parameters for other tasks which is also referred to as end-to-end training.
>
>
> (Continued below)

---

> > ### Author Response · Authors · 2018-11-15
> > **(continuation of ) Author Response for AnonReviewer2**
> >
> > 3) “The authors could evaluate their model with respect to pixel loss (after ground-truth rendering) and compare to a video prediction algorithm (such as PredNet by Lotter, Kreiman, & Cox, 2016).”
> >         The goal of our work was to investigate whether real-world data can be used to learn models of physics and also simultaneously estimate physical parameters in real-world scenes. We do not, however, deal with the realistic rendering of the predicted outputs from the learned physics model. Therefore, we cannot directly compare to future-prediction models like PredNet [Lotter et al], since we do not predict the pixels in the future frames.
> >
> > 4) “I would love to see training curves with errorbars of the models on the most important metrics (e.g. Dist and COR Median Absolute Error)”
> >         We have now computed the error bars for the Forward prediction distance error and COR Median absolute error over multiple training/testing runs with different initializations. These results confirm the conclusions of our ablative study.
> > Experiment			Dist (Mean, Std)		COR Med Abs Err (Mean, Std)
> > Center based			28.2, 0.005			0.173, 0.01
> > Fix core and traj. enc. 	        38.4, 0.008			0.258, 0.008
> > Train core and traj. Enc.	24.7, 0.004			0.169, 0.006
> > Train core, Fix traj. Enc.	21.9, 0.006			0.158, 0.01
> >
> >
> > Clarifications:
> > 1) “How was the PointNet trajectory encoder trained? Were gradients passed through from the PIM?  Was the same network used for both the simulation and real-world data?”
> >         Yes, the PointNet trajectory encoder is actually part of the PIM in our proposed approach. The gradients for the trajectory encoder are computed with respect to the objectives mentioned in Equations (2) and (3).
> > Yes, the same network is used for simulation and real-world data.
> >
> > 2) “The performance of the center-based model in Table 1 seems surprisingly low. Is the VIM at fault?  Or is the sphere-fitting sub-optimal?”
> >         In theory, if accurate centers and point clouds are available, both models should perform similarly. The sphere-fitting in our data is sub-optimal due to the noise in the stereo-depth estimates. We believe that this highlights the advantage of using a PointNet-based model to avoid dealing with hand-crafted estimates of centers.
> >
> > [a] Jui-Hsien Wang, Rajsekhar Setaluri, Dinesh K. Pai, and Doug L. James. Bounce maps: An improved restitution model for real-time rigid-body impact. ACM Transactions on Graphics (Proceedings of SIGGRAPH 2017), 36(4), July 2017. doi: https://doi.org/10.1145/3072959.3073634.

---

> > > ### Comment · AnonReviewer2 · 2018-11-27
> > > **Difficult to Evaluate and Lack of Generality**
> > >
> > > Dear Authors,
> > >
> > > Thank you for your reply.  However, many of my previous comments still apply (also, the paper itself looks to have not been revised much if at all).
> > >
> > > Specifically, my main concerns are these:
> > >
> > > 1)  No comparison to existing baselines.  There are many baselines against which you could compare.  For example, you can compare to video prediction baselines on the simulation data (where you can use the simulation renderer to render trajectories).  Also on the simulated data you could compare to state-based prediction models, such as (Battaglia et al., 2016) that you reference.  Ultimately, as a reader I have no idea how well your model actually models physics.  Given some of the trajectories in Figure 12 it is clear that the model does in fact make mistakes, so this must be compared to existing baselines (even if they don't use exactly the same training paradigm) to verify that it is actually learning the physics well.
> > >
> > > 2)  Lots of hand-holding, lack of generality.  Giving the ground-truth bounce position on the real dataset is a serious assumption.  For more general data, this could be a highly non-trivial preprocessing step and limits the generality of the model.  Similarly for ground-truth knowledge of the impact time.
> > >
> > > Also, a few of my minor comments remain unresolved:
> > > 1) Confidence intervals should be in the paper itself in Tables 1 and 2, preferably as 90% or 95% confidence intervals.
> > > 2) Training curves should be plotted (at least in the supplementary material) corresponding the the tables would be good to see.  The shape of the training curves would indicate how fast the model learns and whether the fine-tuning asymptotes or results in seed-dependent instability (which is common for fine-tuning physics prediction models).
> > >
> > > Stepping back a bit, this paper addresses a very niche problem, because the paradigm involves:
> > > 1) A large simulated dataset and a small real-world dataset (not niche)
> > > 2) Ground-truth impact locations yet no other physical parameters for the real-world dataset (very niche)
> > > 3) Ground-truth knowledge of when the impact occurs in time, and specificity of the model to this time-point (very niche)
> > > 4) Your aim is to infer some unknown physical parameters without actually be able to do rollouts or video prediction (somewhat niche)
> > > 5) The only environment is a single object bouncing (very niche).
> > >
> > > Your model is also very specific to this particular paradigm.  So without strong results (which, given no benchmarking with existing methods, the reader cannot evaluate) I'm struggling to see how this paper could be of interest to the wider ICLR audience.
> > >
> > > While I appreciate your reply, I cannot in good conscience give a rating higher than 5.

---

> > > > ### Author Response · Authors · 2018-11-30
> > > > **Relaxed impact location assumption, Comparison to Battaglia et al. and discussion**
> > > >
> > > > We thank the reviewer for their response and useful suggestions. We have conducted additional experiments that specifically address the major concerns of the reviewer. We compare the PIM model to Interaction Networks (Battaglia et al., 2016) and also relax the assumption of knowledge of impact spatial location.  We show that our PIM model significantly outperforms Interaction Networks on synthetic data and that a model trained without knowledge of ground truth spatial location of impact performs as well as our proposed model. First, we would like to address the high-level concerns of the reviewer.
> > > >
> > > > In the context of applications, it is true that our proposed model can be applied to a single ball collision under knowledge of impact time. However, as a research problem, we believe that this setting and dataset exposes numerous important challenges that currently hinder progress in this direction (discussed in the Introduction). In fact, we believe that what seems like a niche problem is actually a challenging unaddressed elementary problem in modeling real-world collisions.
> > > >
> > > >
> > > > We now present experimental results to address the mentioned concerns:
> > > > 1)  “Also on the simulated data you could compare to state-based prediction models, such as (Battaglia et al., 2016) that you reference.”
> > > >              Thank you for the suggestion. Comparing PIM to Interaction Networks (IN) is indeed an interesting experiment. We have used the simulation data from our experiments to train two versions of the IN model (IN-positions and IN-positions-velocity, described below) using the available codebase. Forward prediction error at 0.1s post-bounce (“Dist.” in the main text) from the simulation-based PIM (as described in “Pretraining the PIM” in Section 3.1) and the two IN Models are as follows:
> > > >
> > > > Center-based PIM: 11.72cm (stdev: 0.009)
> > > > PointNet-based PIM: 12.87cm (stdev: 0.005)
> > > > IN-position-velocity: 36.14cm (stdev: 0.023)
> > > > IN-pos1-pos2-pos3: 23.22cm (stdev: 0.015)
> > > >
> > > > We observe that the PIM model performs significantly better than the Interaction Networks models due to the iterative nature of IN which leads to compounding of errors over time. The Center-based and PointNet-based PIM models perform similarly on the simulation data, but the PointNet-based model is more robust to sensor noise on real data as shown in Table 1.
> > > >
> > > > IN-position-velocity: State vector of the object at t=1 contains [x, y, z, v_x, v_y, v_z]  (used in the original Interaction Networks paper)
> > > > IN-pos1-pos2-pos3: State vector of the object at t=3 contains [x1, y1, z1, x2, y2, z2, x3, y3, z3]
> > > >
> > > >
> > > > Results continued below

---

> > > > > ### Author Response · Authors · 2018-11-30
> > > > > **(cont.) Relaxed impact location assumption, Comparison to Battaglia et al. and discussion**
> > > > >
> > > > >
> > > > > 2)  “Lots of hand-holding, lack of generality.  Giving the ground-truth bounce position on the real dataset is a serious assumption.”
> > > > >                  While we make the assumption of knowing the impact spatial location, it can be automatically estimated. We demonstrate this by using the RANSAC-estimated point cloud center in the collision frame as the collision point and retraining the best model (Row 5 Table 1: “Train core, Fix traj. enc.”).
> > > > > 						           Dist.			% Normals		COR Median Abs Err
> > > > > 					               (Mean, Std)	        (Mean, Std)		       (Mean, Std)
> > > > > Known Impact Loc 		       21.9, 0.006	 	27.06, 0.09		        0.158, 0.01
> > > > > Auto. Estimated Impact Loc .   21.7, 0.009		27.58, 0.04		        0.153, 0.01
> > > > >
> > > > > The assumption of impact location simply allowed us to create the experimental setup to investigate the main goal of our work: estimation of physical parameters by learning from observed bounces.
> > > > >
> > > > >
> > > > >  “Confidence intervals should be in the paper itself in Tables 1 and 2”
> > > > >  “Training curves should be plotted”
> > > > >            We will add these to the final version of the paper, if accepted. While the PDF revision deadline has passed, we are hoping to incorporate all discussions with the reviewers.
> > > > >
> > > > >
> > > > >
> > > > > Discussion about the problem:
> > > > > Point 1: We are agreed “not niche”.
> > > > > Point 2: We hope this concern is now addressed by our removal of the assumption of knowing the spatial location of the impact (in the experiment above) by training with automatically estimated bounce positions.
> > > > >
> > > > > Points 3 and 4: Collision observation, detection and simulation of the non-rigid surfaces that are encountered in everyday scenes remains a challenging task[a] and the subject of active research. Further, in our setting, we have the added challenge of approximate and noisy estimates of the scene geometry; collision detection and processing with uncertainty is just recently considered [b] and there are no standardized codes or methods in the deformable setting. We show post-bounce predictions in Figures 1 and 3 and will release videos of our post-bounce predictions (ICLR’s openreview did not have a way for us to submit the videos anonymously as supplemental material). Note that rollouts require collision detection, which is challenging as previously noted. Collision detection and rollouts are interesting for future work.
> > > > >
> > > > > Point 5: As discussed in the first paragraph in the introduction, modeling single-object bounces has potential application in augmented reality for dynamic object compositing. Handling multiple interacting objects is interesting future work, but the single object setting is a first and necessary step towards it.
> > > > >
> > > > > [a] Collision Detection for Deformable Objects. M. Teschner, S. Kimmerle, G Zachmann, B. Heidelberger, L Raghupathi, A. Fuhrmann, M-P Cani, F Faure, N. Magnenat-Thalmann, and W. Strasser. Eurographics 2004, State-of-the-Art Report
> > > > > [b] Fast and Bounded Probabilistic Collision Detection for High-DOF Trajectory Planning in Dynamic Environments. C. Park, J.S. Park, and D. Manocha. IEEE Transactions on Automation Science and Engineering. 2018.

---

### Meta-Review · Area_Chair1 · 2018-12-14

**Confidence:** 4
**Recommendation:** Accept (Poster)

**Metareview:**

This paper proposes a novel dataset of bouncing balls and a way to learn the dynamics of the balls when colliding. The reviewers found the paper well-written, tackling an interesting and hard problem in a novel way. The main concern (that I share with one of the reviewers) is about the fact that the paper proposes both a new dataset/environment *and* a solution for the problem. This made it difficult the for the authors to provide baselines to compare to.  The ensuing back and forth had the authors relax some of the assumptions from the environment and made it possible to evaluate with interaction nets.

The main weakness of the paper is the relatively contrived setup that the authors have come up with. I will summarize some of the discussion that happened as a result of this point: it is relatively difficult to see how this setup that the authors have and have studied (esp. knowing the groundtruth impact locations and the timing of the impact) can generalize outside of the proposed approach. There is some concern that the comparison with interaction nets was not entirely fair.

I would recommend the authors redo the comparisons with interaction nets in a careful way, with the right ablations, and understand if the methods have access to the same input data (e.g. are interaction nets provided with the bounce location?).

Despite the relatively high average score, I think of this paper as quite borderline, specifically because of the issues related to the setup being too niche. Nonetheless, the work does have a lot of scientific value to it, in addition to a new simulation environment/dataset that other researchers can then use. Assuming the baselines are done in a way that is trustworthy, the ablation experiments and discussion will be something interesting to the ICLR community.